# FM-CLIP: Flexible Modal CLIP for Face Anti-Spoofing

Ajian Liu*
Chinese Academy of Sciences
Beijing, China
ajian.liu@ia.ac.cn

Hui Ma*
Macau University of Science and Technology
Macau
3220006153@student.must.edu.mo

Junze Zheng
Macau University of Science and Technology
Macau
3230006102@student.must.edu.mo

Haocheng Yuan
Macau University of Science and Technology
Macau
1210027647@student.must.edu.mo

Xiaoyuan Yu
Macau University of Science and Technology
Macau
acreateryxy@gmail.com

Yanyan Liang†
Macau University of Science and Technology
Macau
yyliang@must.edu.mo

Sergio Escalera
University of Barcelona
Barcelona, Spain
sergio.escalera.guerrero.gmail.com

Jun Wan
Chinese Academy of Sciences
Beijing, China
jun.wan@ia.ac.cn

Zhen Lei
MAIS, CASIA
School of Artificial Intelligence, UCAS
CAIR, HKISI, CAS
zlei@nlpr.ia.ac.cn

## ABSTRACT

Flexible modal Face Anti-spoofing (FAS) aims to aggregate all the available training modalities' data to train a model and enables flexible testing of any given modal samples. In this work, borrowing a solution from the large-scale vision-language models (VLMs) instead of directly removing modality-specific signals from visual features, we propose a novel Flexible Modal CLIP (**FM-CLIP**) for flexible modal FAS, that can utilize text features to dynamically adjust visual features to be modality independent. In the visual branch, considering the huge visual differences of the same attack in different modalities, which makes it difficult for classifiers to flexibly identify subtle spoofing clues in different test modalities, we propose Cross-Modal Spoofing Enhancer (**CMS-Enhancer**). It includes a Frequency Extractor (**FE**) and Cross-Modal Interactor (**CMI**), aiming to map different modal attacks in a shared frequency space to reduce interference from modality-specific signals and enhance spoofing clues by leveraging cross-modal learning from the shared frequency space. In the text branch, we introduce a Language-Guided Patch Alignment (**LGPA**) based on prompt learning, which further guides the image encoder to focus on patch-level spoofing representations through dynamic weighting by text features. Thus, our FM-CLIP can flexibly test different modal samples by identifying and enhancing modality-agnostic spoofing cues. Finally,

extensive experiments show that FM-CLIP is effective and outperforms state-of-the-art methods on multiple multi-modal datasets.

## CCS CONCEPTS

• **Computing methodologies** → **Computer vision**.

## KEYWORDS

Flexible modal Face Anti-spoofing, CLIP, Cross-Modal Spoofing Enhancer, Frequency Extractor, Cross-Modal Interactor, Language-Guided Patch Alignment

**ACM Reference Format:**
Ajian Liu*, Hui Ma*, Junze Zheng, Haocheng Yuan, Xiaoyuan Yu, Yanyan Liang†, Sergio Escalera, Jun Wan, and Zhen Lei. 2024. FM-CLIP: Flexible Modal CLIP for Face Anti-Spoofing. In *Proceedings of Make sure to enter the correct conference title from your rights confirmation email (MM'24)Proceedings of the 32nd ACM International Conference on Multimedia (MM'24), October 28-November 1, 2024, Melbourne, Australia.* ACM, New York, NY, USA, 10 pages. https://doi.org/10.1145/3664647.3680856

## 1 INTRODUCTION

The task of Face Anti-spoofing (FAS) is to protect face recognition systems from physical media-based presentation attacks, such as print [53, 54], replay [3] and mask [4, 21]. With the increasing advancement of attack mediums and spoofing methods, FAS algorithms [14, 20, 29, 30, 50, 51] designed based on RGB modality have become challenging for realistic spoofing clues. The multi-modal algorithms [8, 9, 17, 53] target these clues by leveraging the complementary advantages of multi-modal samples, which insist that the same attack is difficult to evade detection from multiple spectra simultaneously. However, these multi-modal fusion methods need to provide samples of the same modality during training and testing. When any modality disappears during testing, these methods will not be able to differentiate between real and fake faces, resulting in poor performance. Due to hardware cost and space constraints, it

* Equal contributions.
† Corresponding Author.

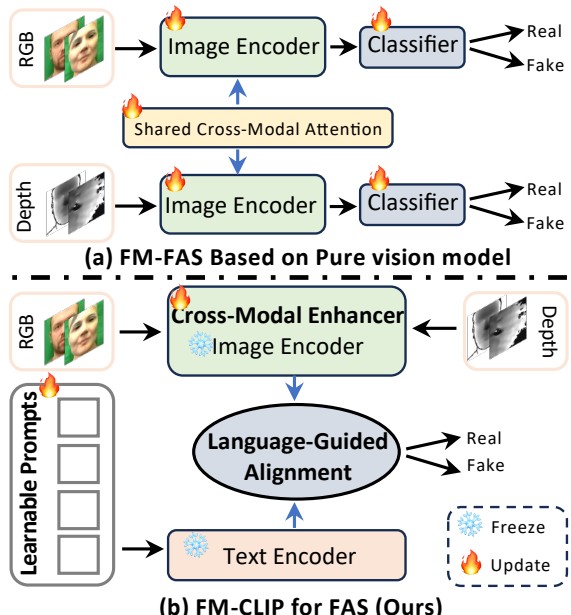

**Figure 1: (a) The pure visual encoder scheme for flexible modal tasks requires setting up an image encoder independently for each modality sample to learn modality-related features and establishing a shared cross-modal attention mechanism between them to learn modality-agnostic features. (b) FM-CLIP avoids redundant network design by inserting cross-modal enhancers in the visual branch for cross-modal mutual learning to enhance feature generalization. At the same time, in the text branch, introduce a Language-Guided Alignment based on the prompt learning to focus on spoofing cues through dynamic weighting by text features.**

is not always possible to provide consistent modal samples in practical applications, which makes these systems difficult to deploy widely. Recently, flexible modality FAS [16, 19, 47, 48] has attracted widespread discussion in the community because it can be flexibly deployed in any given test modality environment without providing the same modal type as in the training stage. A type method in this field is FM-ViT [19], which sets up an independent encoder for each modality. After learning individual modality features from different branches, a shared cross-modal block is established between them to guide each branch to learn potential and modality agnostic active features by summarizing multi-modal information as shown in Fig. 1 (a). Due to the significant differences in the same attack type in different modalities, classifiers are highly susceptible to interference from modality-related information.

Inspired by *CLIP* [36], in this work, borrowing a solution from the large-scale vision-language models (VLMs) instead of directly removing modality-specific signals from visual features, we propose a novel Flexible Modal CLIP (**FM-CLIP**) for flexible modal FAS as shown in Fig. 1 (b), that can utilize text features to dynamically adjust visual features to be modality independent. In the visual branch, considering the huge spatial domain differences of the same attack in different modalities, which makes it difficult

for classifiers to flexibly identify subtle spoofing clues in different test modalities, we propose Cross-Modal Spoofing Enhancer (**CMS-Enhancer**). Specifically, it includes a Frequency Extractor (**FE**) and Cross-Modal Interactor (**CMI**). FE converts visual features into the frequency domain through discrete cosine transform (DCT) and further adaptively adjusts the distribution of spatial domain features to mine essential spoofing clues. CMI conducts cross-modal mutual learning of effective cues in each modality to enhance the features of a single modality. FE and CMI work together to map different modal attacks in a shared frequency space to reduce interference from modality-specific signals and enhance spoofing clues by leveraging cross-modal learning from the shared frequency space. In the text branch, considering the alignment between image features and text feature embeddings can lead to non-trivial generalization improvements, so we introduce a Language-Guided Patch Alignment (**LGPA**) based on the prompt learning, which further guides the image encoder to focus on patch level spoofing representations through dynamic weighting by text features. Specifically, we follow CoOp [57] initializes learnable content embeddings, which are fed into a text encoder along with handcrafted categories to produce text supervision signals. The LGPA aligns the local image patch tokens with global text prompt embeddings. Finally, we combine visual patch-based alignment and visual CLS token-based alignment to supervise model training. The main contributions of this paper are summarized as follows:

- We borrowed CLIP to propose a Flexible Modal CLIP (FM-CLIP) for FAS, which can utilize text features to dynamically adjust visual features to make them independent of modality.
- In the visual branch, we propose Cross-Modal Spoofing Enhancer (CMS-Enhancer), including Frequency Extractors (FE) and Cross-Modal Interactors (CMI), aiming to map different modal attacks into a shared frequency space to perform cross-modal learning so that the model can effectively pay attention to subtle spoofing cues.
- In the text branch, we introduce language-guided patch alignment (LGPA) based on prompt learning, which further guides the image encoder to focus on patch-level spoofing representation through dynamic weighting of text features.
- Extensive experiments demonstrate FM-CLIP's effectiveness and superior performance over state-of-the-art results on WMCA, CASIA-SURF, and CASIA-SURF CeFA datasets.

## 2 RELATED WORK

**Single-Modal FAS Methods.** With the rise of deep learning frameworks, Some CNN-based methods [13, 31] design a unified framework for feature extraction and classification in an end-to-end manner. Another works [29, 38, 45, 51] utilize physical-based depth information as a supervisory signal instead of binary classification loss, improving the model's ability to perceive and understand scene depth more effectively. Although these algorithms have achieved astonishing results in intra-datasets experiments, their performance deteriorates severely when faced with unknown domains.

To solve these limits, domain generalization-based FAS is increasingly receiving attention from researchers. Some Domain Adaptation-based methods [25, 26, 42, 44, 52] aim to minimize the distribution discrepancy between the source and target domain

by leveraging the unlabeled target data. Another Domain Generalization based methods [10, 14, 20, 23, 24, 37, 39–41, 43, 50] can conquer this by taking the advantage of multiple source domains. MFAE [56] randomly masks the low-frequency spectrum of images and reconstructs the images for self-supervised pre-training of Vision Transformers, and also integrates an auxiliary content-regularization decoder to further enhance the model's insensitivity to low-frequency features. CIFAS [27] addresses the challenge of domain bias by introducing causal intervention techniques. Liu et al. [28] propose UDG-FAS, the first Unsupervised DG framework for FAS. CFPL-FAS [20] targets DG FAS via textual prompt learning for the first time, which utilizes two lightweight models to learn the different semantic prompts conditioned on content and style features. CA-MoEiT [14] introduces professional experts and super expert to solve DG FAS.

**Multi-Modal Fusion Methods.** Multi-modal FAS has gained attention due to the increasing sophistication of high-quality 2D attacks, such as those present in datasets like OULU-NPU [1], SiW [29], CelebA-Spoof [55] as well as high-fidelity mask attacks, including MARsV2 [22], and HiFiMask [21]. These attacks exhibit realistic color, and texture making it challenging to detect spoofing clues using only the visible spectrum. To address this issue, multi-modal fusion methods [8, 9, 17, 18, 53] have proven effective by leveraging different modalities that may reveal distinct properties of fake faces. However, these multi-modal fusion-based algorithms require the testing phase to provide the same modal types as the training phase, limiting their deployment scenarios.

**Flexible-Modal Methods.** To alleviate the limitation of consistency between testing and training modalities, flexible modality-based methods [8, 12, 16, 19, 47, 48] aim to improve the performance of any single modality by leveraging available multi-modal data. George et al. [8] presents a framework for PAD that uses RGB and depth channels supervised by the proposed cross-modal focal loss (CMFL), which makes it possible to train models using all the available channels and to deploy with a subset of channels. MA-ViT [16] adopts the early fusion to aggregate all the available training modalities' data and enables flexible testing of any given modal samples with a Modality-Agnostic Transformer Block. Yu et al. [47, 48] introduce a flexible-modal benchmark aimed at training a unified model capable of deployment across various modality scenarios. FM-ViT [19] retains a specific branch for each modality to capture different modal information and introduces the Cross-Modal Transformer Block, which consists of two cascaded attentions named Multi-headed Mutual-Attention and Fusion-Attention. MMDG [12] proposes a multi-modal domain generalization framework MMDG, which addresses modal unreliability and imbalance issues through uncertainty-guided U-Adapter and ReGrad strategy.

## 3 METHOD

**CLIP.** We adopt CLIP [36] as the pre-trained model with a ViT image encoder and transformer text encoder, respectively. Given an image $x \in \mathbb{R}^{3 \times H \times W}$, the vision encoder converts it into a $D$-dimensional image feature $f_{img} \in \mathbb{R}^{(1+N) \times D}$, where 1 represents the CLS token and $N$ denotes the patch tokens. Meanwhile, the text encoder $g(.)$ takes text descriptions $t$ and generates the text feature $f_{text} \in \mathbb{R}^{K \times D}$ from the appended EOS tokens, where $K$ denotes

the number of classes. Formally, let cls token $f_{img}^{(0)}$ extracted for a testing image x, and given the text prompts $P$, we have the predicted similarity of class $i \in \{0, 1\}$, where 0 represents 'real' and 1 is 'fake':

$$p(y \mid x) = \frac{\exp\left(< g(P_y), f_{img}^{(0)} > / \tau\right)}{\sum_{j=1}^{K} \exp\left(< g(P_j), f_{img}^{(0)} > / \tau\right)}, \quad (1)$$

where $\tau$ is a temperature coefficient, and y denotes the label.

### 3.1 Cross-Modal Spoofing Enhancer

To cope with the difficulty of classifiers in identifying subtle spoofing clues caused by the huge difference in the representation of the same attack type in different modalities, we introduce a Cross-Modal Spoofing Enhancer (CMS-Enhancer) to bridge the adjacent ViT stage in the image encoder as shown in Fig. 2. CMS-Enhancer divides sample features into spatial and frequency domain features and allows cross-modal interactive learning of frequency domain features of different modalities in a shared frequency domain band to enhance the model's capture of spoofing clues.

**Spatial Extractor (SE).** Benefiting from the hard-coded inductive bias of convolutional layers, inserting lightweight convolutional layers in the ViT encoder is more suitable for vision tasks. Inspired by this, we construct a lightweight Spatial Extractor (SE) in the spatial domain, comprising three convolution layers and two GELU layers for capturing subtle spoofing clues, as follows

$$F_{SE\_output}^{(j)} = \text{Conv1}\left(\text{GELU}\left(\text{Conv3}\left(\text{GELU}\left(\text{Conv1}\left(F_{input}^{(j)}\right)\right)\right)\right)\right), \quad (2)$$

$$\hat{F}_{spatial}^{(j)} = F_{SE\_output}^{(j)} \oplus F_{input}^{(j)}, \quad (3)$$

where Conv1 and Conv3 represent the 1x1 and 3x3 convolution kernel respectively; $F_{input}^{(j)}$ and $F_{SE\_output}^{(j)}$ represent the vanilla transformer layer features and output features extracted by the SE module in j-th ViT stage respectively; $\hat{F}_{spatial}^{(j)}$ represents the spatial feature; $\oplus$ represents element-wise summation.

**Frequency Extractor (FE).** As is known, RGB images and depth maps contain various information. Depth maps contain more contours, while RGB images convey rich detailed information, such as texture and color. Huge differences in representations of different modalities can cause classifiers to overly focus on modal content information. In the frequency domain, an image is represented as a superposition of its different frequency components. Inspired by FcaNet [35], we construct a lightweight Frequency Extractor (FE) in the frequency domain, comprising Discrete Cosine Transform (DCT), two convolution layers, and one GELU layers, as follows

$$F_{FE\_output}^{(j)} = \sigma\left(\text{Conv1}\left(\text{GELU}\left(\text{Conv1}\left(\text{DCT}\left(F_{input}^{(j)}\right)\right)\right)\right)\right), \quad (4)$$

$$\hat{F}_{frequency}^{(j)} = F_{FE\_output}^{(j)} \otimes F_{input}^{(j)}, \quad (5)$$

where $\sigma$ is the sigmoid function, and $\otimes$ represents element-wise multiplication. The 2D DCT is mathematically defined as follows:

$$\text{DCT}(F) = \sum_{x=0}^{H-1} \sum_{y=0}^{W-1} F * \cos\left(\frac{\pi h}{H}\left(x + \frac{1}{2}\right)\right) \cos\left(\frac{\pi w}{W}\left(y + \frac{1}{2}\right)\right),$$

$$h \in \{0, 1, \cdots, H-1\}, w \in \{0, 1, \cdots, W-1\}, \quad (6)$$

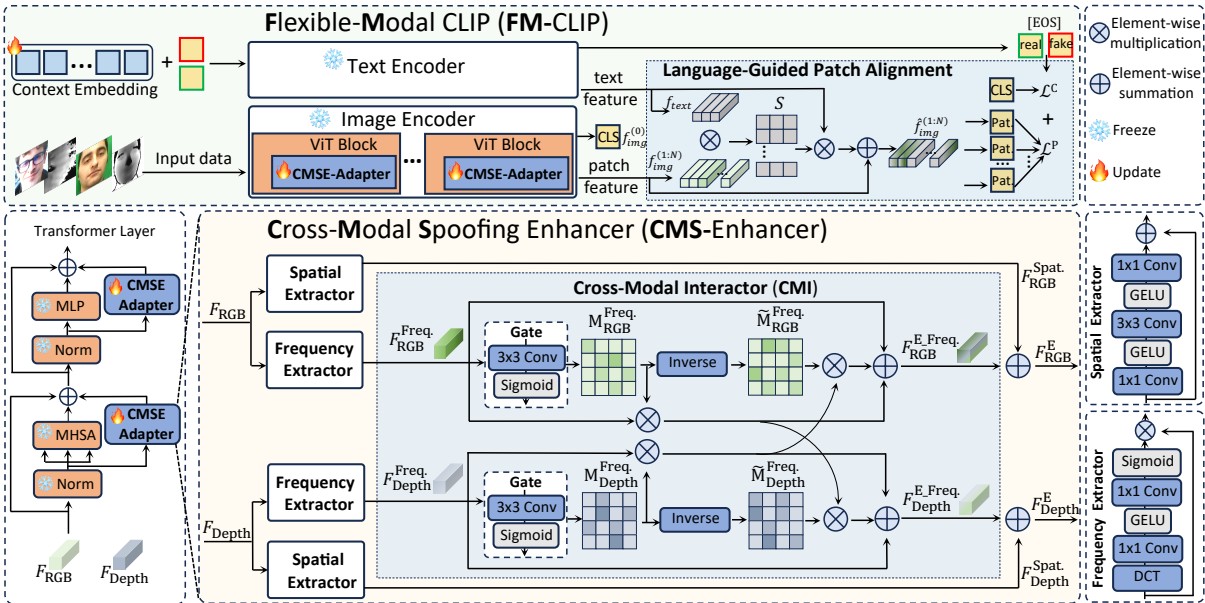

**Figure 2: An overview of the FM-CLIP Framework is based on the frozen CLIP model. In the visual branch, the ViT image encoder integrates the CMS-Enhancer. Taking RGB as an example, use RGB features $F_{RGB}$ as input to obtain $F_{RGB}^{Spat.}$ and $F_{RGB}^{Freq.}$ through Spatial Extractor and Frequency Extractor respectively. Using $F_{RGB}^{Freq.}$ and $F_{Depth}^{Freq.}$ as input through the CMI module, RGB and Depth can be complementary learned in the frequency domain to obtain enhanced features $\mathbf{F}_{RGB}^{E}$ and $\mathbf{F}_{Depth}^{E}$. The final Image Encoder outputs CLS token $f_{img}^{(0)}$ and Patch token $f_{img}^{(1:N)}$. In the text branch, The learnable content vectors and manual category vectors are used as input together with Text Encoder to extract text features $f_{text}$. The $f_{text}$ and The $f_{img}^{(1:N)}$ are input to the LGPA module together, $f_{text}$ guides the patch token $f_{img}^{(1:N)}$ to calculate the similarity matrix $S$, and further obtains $\hat{f}_{img}^{(1:N)}$. The EOS (real, fake) in the last dimension of the text feature $f_{text}$ performs cosine similarity with CLS token $f_{img}^{(0)}$ and Patch token $f_{img}^{(1:N)}$ respectively, and then uses cross-entropy function with label $y \in \{0, 1\}$ to calculate loss $\mathcal{L}^P$ and $\mathcal{L}^C$.**

where H and W are the height and width of the input feature map. Different modal features are mapped in a shared frequency domain, and the Cross-Modal Interactor module is used to complementarily enhance the spoofing-related representation of each modality.

**Cross-Modal Interactor (CMI).** Inspired by [46] and [5], we use a gate network to extract discriminative information from each modal feature. Specifically, the input frequency feature $\hat{F}_{frequency}^{(j)}$ is fed into a gate consisting of a 3x3 convolutional layer and a Sigmoid activation function, and then we can obtain the value probability gate maps $\mathbf{M}_{RGB}^{Freq.} \in [0, 1]$ and $\mathbf{M}_{Depth}^{Freq.} \in [0, 1]$. Since the calculation of each ViT stage is the same, in the subsequent formulas, we omit the ViT stage mark ($j$). The formulation is

$$\mathbf{M}_{RGB}^{Freq.} = \sigma\left(Conv3\left(\mathbf{F}_{RGB}^{Freq.}\right)\right); \quad \mathbf{M}_{Depth}^{Freq.} = \sigma\left(Conv3\left(\mathbf{F}_{Depth}^{Freq.}\right)\right), \tag{7}$$

The larger the value in this map, the more discriminative information the feature vectors in $F_{RGB}^{Freq.}$ have at their corresponding positions. On the contrary, the smaller the value, the smaller the amount of valuable information. Furthermore, the uninformative

gate maps can be represented as $\widetilde{\mathbf{M}}_{RGB}^{Freq.}$ and $\widetilde{\mathbf{M}}_{Depth}^{Freq.}$ as follows:

$$\widetilde{\mathbf{M}}_{RGB}^{Freq.} = \mathbf{1} - \mathbf{M}_{RGB}^{Freq.}; \quad \widetilde{\mathbf{M}}_{Depth}^{Freq.} = \mathbf{1} - \mathbf{M}_{Depth}^{Freq.} \quad , \tag{8}$$

Two unimodal features $\widetilde{\mathbf{F}}_{RGB}^{Freq.}$ and $\widetilde{\mathbf{F}}_{Depth}^{Freq.}$ are obtained by:

$$\widetilde{\mathbf{F}}_{RGB}^{Freq.} = \mathbf{M}_{RGB}^{Freq.} \otimes \mathbf{F}_{RGB}^{Freq.}; \quad \widetilde{\mathbf{F}}_{Depth}^{Freq.} = \mathbf{M}_{Depth}^{Freq.} \otimes \mathbf{F}_{Depth}^{Freq.} \quad , \tag{9}$$

Through such calculations, the model can effectively learn useful information from features, thereby suppressing the interference of useless information on the model. To utilize the cross-modal complementary relationships, we perform the same operation between the uninformative gate maps ($\widetilde{\mathbf{M}}_{RGB}^{Freq.}$, $\widetilde{\mathbf{M}}_{Depth}^{Freq.}$) and discriminatory features from another modality ($\widetilde{\mathbf{F}}_{RGB}^{Freq.}$, $\widetilde{\mathbf{F}}_{Depth}^{Freq.}$) to get complimentary cross-modal features:

$$\widetilde{\mathbf{F}}_{RGB\_Depth}^{Freq.} = \widetilde{\mathbf{M}}_{RGB}^{Freq.} \otimes \widetilde{\mathbf{F}}_{Depth}^{Freq.}; \quad \widetilde{\mathbf{F}}_{Depth\_RGB}^{Freq.} = \widetilde{\mathbf{M}}_{Depth}^{Freq.} \otimes \widetilde{\mathbf{F}}_{RGB}^{Freq.}, \tag{10}$$

Please note that the complementary information from the deep modality is selected by the uninformative gate map of the RGB

modality, which means it propagates to positions with little information in the RGB features. In addition, we adopt residual connections to preserve the original feature of each modality. Therefore, we can capture the complementary features across modalities. So we can obtain these two features in the following way:

$$
\begin{cases}
\mathbf{F}_{RGB}^{E\_Freq.} = \mathbf{F}_{RGB}^{Freq.} \oplus \widetilde{\mathbf{F}}_{RGB}^{Freq.} \oplus \widetilde{\mathbf{F}}_{RGB\_Depth}^{Freq.} & , \\
\mathbf{F}_{Depth}^{E\_Freq.} = \mathbf{F}_{Depth}^{Freq.} \oplus \widetilde{\mathbf{F}}_{Depth}^{Freq.} \oplus \widetilde{\mathbf{F}}_{Depth\_RGB}^{Freq.} & ,
\end{cases}
\tag{11}
$$

Furthermore, the frequency domain enhanced features and spatial domain features corresponding to each modal sample are fused

$$
\mathbf{F}_{RGB}^{E} = \mathbf{F}_{RGB}^{E\_Freq.} \oplus \mathbf{F}_{RGB}^{Spat.}; \quad \mathbf{F}_{Depth}^{E} = \mathbf{F}_{Depth}^{E\_Freq.} \oplus \mathbf{F}_{Depth}^{Spat.},
\tag{12}
$$

where $\mathbf{F}_{RGB}^{E}$ and $\mathbf{F}_{Depth}^{E}$ represent enhanced features; $\mathbf{F}_{RGB}^{Spat.}$ and $\mathbf{F}_{Depth}^{Spat.}$ represent the spatial feature extracted by SE model.

## 3.2 Vision-Language Alignment

In the text branch, we introduce a Language-Guided Patch Alignment based on prompt learning, which further guides the image encoder to focus on patch-level spoofing cues through dynamic weighting by text features. In a bit more detail, prompt learning initializes learnable content embeddings, which are fed into a text encoder along with handcrafted categories to produce text supervision signals. The LGPA aligns image patches with text embedding features. Finally, we combine patch-based and CLS token-based alignment to supervise model training.

**Prompt learning.** We follow CoOp [57] to avoid prompt engineering to further enhance the migration capabilities of CLIP models. Different from the zero-shot transfer that used a fixed hand-craft prompt, we construct and fine-tune a set of M continuous context vectors $\boldsymbol{v} = \{\boldsymbol{v}_1, \boldsymbol{v}_2, ..., \boldsymbol{v}_M\}$ as the turntable prompt. Specifically, the prompt $\boldsymbol{t}_i = \{\boldsymbol{v}_1, \boldsymbol{v}_2, ..., \boldsymbol{v}_M, \boldsymbol{c}_i\}$ combines the learnable context vectors $\boldsymbol{v}$ and the class embedding $\boldsymbol{c}_i$, and is fed to the text encoder $g(\cdot)$. The probability for $y$-th class is obtained as

$$
p_{cls\_token}(y \mid \boldsymbol{x}) = \frac{\exp\left(< g\left(t_y\right), f_{img}^{(0)} > /\tau\right)}{\sum_{j=1}^{K} \exp\left(< g\left(t_j\right), f_{img}^{(0)} > /\tau\right)},
\tag{13}
$$

**Language-Guided Patch Alignment (LGPA).** We propose a Language-Guided Patch Alignment, which aligns the image patch with text features to guide visual networks to attend to spoofing cues adaptively inspired by [33]. Specifically, given the text prompt embeddings $f_{text} \in \mathbb{R}^{K \times D}$ from text encoder and image patch tokens $f_{img}^{(1:N)} \in \mathbb{R}^{N \times D}$, our Language-guided calculates the similarity matrix $S$ between them by

$$
S = f_{img}^{(1:N)} \cdot (f_{text})^T .
\tag{14}
$$

where $\cdot$ represents matrix multiplication, and $S \in \mathbb{R}^{N \times K}$. We fuse textual features with similar visual representations in image patches,

$$
\hat{f}_{img}^{(1:N)} = \text{softmax}(S) \cdot f_{text} + f_{img}^{(1:N)},
\tag{15}
$$

where $\hat{f}_{img}^{(1:N)}$ denotes the language-guided image patch tokens.

For the loss calculation, we combine visual patch-based alignment and visual CLS token-based alignment to supervise model

training. For visual CLS token-based alignment, we apply Eq. 13 to calculate probability. For visual patch-based alignment, we calculate image patch tokens probability, as follows

$$
p_{patch\_token}(y \mid \boldsymbol{x}) = \frac{1}{N} \sum_{n=1}^{N} \frac{\exp\left(< g\left(t_y\right), \hat{f}_{img}^{(n)} > /\tau\right)}{\sum_{j=1}^{K} \exp\left(< g\left(t_j\right), \hat{f}_{img}^{(n)} > /\tau\right)},
\tag{16}
$$

In this paper, we adopt the cross entropy loss on Eq. 13 and Eq. 16 with label $y \in \{0, 1\}$ to calculate loss, as follows

$$
\mathcal{L} = -y \cdot \log p(y \mid \boldsymbol{x}) - (1-y) \cdot \log(1 - p(y \mid \boldsymbol{x})),
\tag{17}
$$

the total loss is as follows

$$
\mathcal{L}^{total} = \mathcal{L}^{P} + \mathcal{L}^{C} \quad ,
\tag{18}
$$

where $\mathcal{L}^{P}$ and $\mathcal{L}^{C}$ represent patch loss and CLS loss respectively.

## 4 EXPERIMENTS

**Datasets & Protocols.** We use three commonly used multi-modal FAS datasets for experiments, including CASIA-SURF (SURF) [53], CASIA-SURF CeFA (CeFA) [17], WMCA [9]. SURF [53] consists of $1,000$ subjects with $21,000$ videos and each sample has 3 modalities, and we follow a protocol to evaluate the performance against unknown attack types. CeFA [17] covers 3 modalities, $1,607$ subjects, and provides five protocols. We select the Protocols 1, 2, and 4 for experiments. WMCA [9] contains a wide variety of presentation attacks, which introduces 2 protocols: the grandest protocol emulates the "seen" attack scenario and the "unseen" protocol evaluates the generalization of an unseen attack.

**Test Scenario Settings&Evaluation Metrics.** We follow the test scenarios of FM-ViT [19]. The first is a commonly used setting where the test modalities need to be consistent with the training stage. The second is a flexible modal test scenario, which means the user can provide any single-modal sample. Attack Presentation Classification Error Rate (APCER), Bonafide Presentation Classification Error Rate (BPCER), and ACER [11] are used for the metrics.

**Implementation Details.** We utilize ViT-B/16 as an image encoder, whose output embedding dimension is 768. Meanwhile, we employ a transformer-based text encoder. In the training stage, the input images are cropped and resized to 224 x 224 x3. We apply random cropping and random horizontal flipping at training, while center cropping at testing, both with no other augmentations. We use an Adam optimizer with an initial learning rate of 1e-6 and weight decay of 1e-6. We train all methods with a maximum of 200 epochs, and FM-CLIP is trained with a batch size of 12. Unless otherwise stated, all ablation experiments are conducted with WMCA ('seen') data in the flexible modality test scenario.

### 4.1 Experimental Result

**Fixed Modal Scenario Evaluations.** The fixed modal scenario setting evaluates the fusion ability of the FM-CLIP framework approach to multi-modal information. We verified the performance of the CMS-Enhancer and the FM-CLIP overall framework. **(SURF):** As shown in Tab. 1, after adding CMS-Enhancer to the visual network ViT, compared to FM-ViT, the ACER was reduced from 0.45 to 0.44. Further, we added LGPA to form the final FM-CLIP, and the ACER was further reduced to 0.43. VisioinLabs used additional datasets,

**Table 1: The results on SURF. A large TPR(%) and a lower ACER (%) indicate better performance. The best results are bolded.**

| Method | TPR | | | APCER | BPCER | ACER |
|---|---|---|---|---|---|---|
| | @FPR=$10^{-2}$ | @FPR=$10^{-3}$ | @FPR=$10^{-4}$ | | | |
| MS-SEF [53] | 99.70 | 97.40 | 92.40 | 1.90 | **0.10** | 1.00 |
| VisionLabs [34] | **99.98** | **99.95** | **99.87** | **0.01** | 0.15 | **0.08** |
| ViT | 87.58 | 63.09 | 27.05 | 3.94 | 4.48 | 4.21 |
| FM-ViT [19] | 99.83 | 99.13 | 98.23 | 0.39 | 0.50 | 0.45 |
| CMS-Enhancer | 99.84 | 99.14 | 98.22 | 0.41 | 0.47 | 0.44 |
| FM-CLIP | 99.84 | 99.12 | 98.25 | 0.42 | 0.45 | 0.43 |

**Table 2: Comparison of ACER (%) values on Protocol "seen" and "unseen" for the WMCA. The best results are bolded.**

| Method | seen | unseen | | | | | | | |
|---|---|---|---|---|---|---|---|---|---|
| | | Flexiblemask | Replay | Fakehead | Prints | Glasses | Papermask | Rigidmask | Mean±Std |
| MC-PixBiS [6] | 1.80 | 49.70 | 3.70 | 0.70 | 0.10 | 16.00 | **0.20** | 3.40 | 10.50±16.70 |
| MCCNN-OCCL-GMM [7] | 3.30 | 22.80 | 31.40 | 1.90 | 30.00 | 50.00 | 4.80 | 18.30 | 22.74±15.30 |
| CMFL [8] | 1.70 | 12.40 | 1.00 | 2.50 | 0.70 | 33.50 | 1.80 | 1.70 | 7.60±11.20 |
| ViT | 2.71 | 11.95 | 1.44 | 3.78 | 0.00 | 18.02 | 0.58 | 4.43 | 5.74±6.75 |
| FM-ViT [19] | **1.0** | 3.56 | 0.72 | 0.00 | 0.00 | 12.00 | 0.43 | 0.73 | 2.49±4.37 |
| CMS-Enhancer | 1.06 | **3.23** | 0.70 | **0.00** | **0.00** | 11.2 | 0.39 | 0.69 | 2.36±4.08 |
| FM-CLIP | 1.05 | 3.35 | **0.69** | **0.00** | **0.00** | **11.0** | **0.38** | **0.66** | **2.29±4.00** |

**Table 3: Evaluation results (%) on the Protocol 1, 2, and 4 of CeFA dataset.**

| Pro. | Method | APCER(%) | BPCER(%) | ACER(%) |
|---|---|---|---|---|
| 1 | PSMM [17] | 2.40±0.60 | 4.60±2.30 | 3.50±1.30 |
| | ViT | 1.42±0.51 | 1.58±1.88 | 1.50±0.77 |
| | FM-ViT [19] | 1.29±1.21 | 0.67±0.95 | 0.98±0.31 |
| | CMS-Enhancer | 1.25±1.05 | 0.68±1.05 | 0.97±0.32 |
| | FM-CLIP | **1.25±1.03** | **0.66±0.99** | **0.95±0.32** |
| 2 | PSMM [17] | 7.70±9.00 | 3.10±1.60 | 5.40±5.30 |
| | ViT | 2.82±1.20 | 1.25±0.59 | 1.67±0.83 |
| | FM-ViT [19] | **0.46±0.09** | 1.08±0.83 | 0.30±0.07 |
| | CMS-Enhancer | 0.47±0.13 | 1.05±0.79 | 0.33±0.06 |
| | FM-CLIP | 0.47±0.12 | **1.04±0.80** | **0.30±0.06** |
| 4 | PSMM [17] | 7.80±2.90 | 5.50±3.00 | 6.70±2.20 |
| | Hulking [15] | 3.25±1.98 | 1.16±1.12 | 2.21±1.26 |
| | Super [15] | **0.62±0.43** | 2.75±1.50 | 1.68±0.54 |
| | BOBO [49] | 1.05±0.62 | 1.00±0.66 | 1.02±0.59 |
| | ViT | 3.17±2.15 | 6.83±6.08 | 5.00±2.19 |
| | FM-ViT [19] | 0.87±1.16 | 0.93±1.53 | 0.90±1.34 |
| | CMS-Enhancer | 0.79±1.24 | 0.95±1.41 | 0.88±1.31 |
| | FM-CLIP | 0.77±1.28 | **0.93±1.45** | **0.87±1.25** |

and secondly, the CMI in our framework FM-CLIP uses cross-modal complementarity and does not fuse features, so it is inferior to the VisioinLabs method. **(WMCA):** As shown in Tab. 2, Compared with FM-ViT, the average ACER of CMS-Enhancer on the WMCA 'unseen' protocol dropped from 2.49±4.37 to 2.36±4.08. On this basis,

FM-CLIP further dropped to 2.29±4.00. Since the number of trainable parameters of FM-CLIP is 5.34M, which is 17.43M less than FM-ViT's 22.77M, as shown in the Tab. 9, its performance on the WMCA 'seen' protocol is inferior to FM-ViT. **(CeFA):** As shown in Tab. 3, compared to FM-ViT, FM-CLIP has decreased in three indicators: APCER, BPCER, and ACER from 1.29±1.21, 0.67±0.95, and 0.98±0.31 to 1.25±1.03, 0.66±0.99, and 0.95±0.32, respectively, on Protocol 1. Compared to FM-ViT, FM-CLIP has decreased in two indicators: BPCER, and ACER from 1.08±0.83, and 0.30±0.07 to 1.04±0.80, and 0.30±0.06, respectively, on Protocol 2. Compared to FM-ViT, FM-CLIP has decreased in three indicators: APCER, BPCER, and ACER from 0.87±1.16, 0.93±1.53, and 0.90±1.34, to 0.77±1.28, 0.93±1.45, and 0.87±1.25, respectively, on Protocol 4. For fixed modal scenario evaluations, extensive experiments demonstrate FM-CLIP's effectiveness and superior performance over SOTA results on SURF, WMCA, and CeFA datasets.

**Flexible Modal Scenario Evaluations.** We conducted experiments under flexible modal scenarios on SURF, CeFA (Protocol 4), and WMCA (Protocol 'seen') data as shown in Tab. 4. Similarly, we verified the performance of CMS-Enhancer and FM-CLIP Framework respectively. **SURF:** Compared with FM-ViT, the ACER of CMS-Enhancer in RGB, Depth, and IR decreased from 12.38, 3.49, and 2.59 to 10.3, 3.18, and 2.23 respectively. Furthermore, FM-CLIP achieves 10.21, 3.02, and 2.01 and reduces 2.17, 0.47, and 0.58 compared to FM-ViT respectively. **CeFA (Protocol 4):** Compared with FM-ViT, the ACER of CMS-Enhancer in RGB, and IR decreased from 21.06±4.90, and 2.88±2.23 to 14.57±1.41, and 2.59±2.24 respectively. Furthermore, FM-CLIP achieves 11.87±3.14, and 2.07±1.36 and reduces 9.19, and 8.1 compared to FM-ViT respectively. **WMCA (protocol 'seen'):** Compared with FM-ViT, the ACER of CMS-Enhancer in RGB, Depth, and IR decreased from 2.87, 2.32, and 2.13 to 2.23,

**Table 4: Comparison of flexible modal results (%) based on multi-modal datasets. The 'SOTA' means the method with public results on the corresponding dataset. R&D&I indicates the method receives RGB (R), Depth (D), and IR (I) paired samples.**

| Method | Train | Test | SURF | | | CeFA (Protocol 4) | | | WMCA (Protocol "seen") | | |
|---|---|---|---|---|---|---|---|---|---|---|---|
| | | | APCER | BPCER | ACER | APCER | BPCER | ACER | APCER | BPCER | ACER |
| | | | | | | Fixed modal testing | | | | | |
| SOTA [7, 49, 53] | R | R | 40.30 | 1.60 | 21.00 | 9.96±5.41 | 2.08±0.88 | 6.02±2.33 | 65.65 | 0.00 | 32.83 |
| | D | D | 6.00 | 1.20 | 3.60 | 4.29±1.37 | 1.17±0.63 | 2.73±0.97 | 11.77 | 0.31 | 6.04 |
| | I | I | 38.60 | 0.40 | 19.40 | 19.61±15.66 | 0.58±0.38 | 10.10±7.66 | 5.03 | 0.00 | 2.51 |
| ResNet50 [19] | R | R | 23.39 | 22.50 | 22.95 | 25.08±1.46 | 25.06±1.41 | 25.07±1.44 | 6.33 | 13.91 | 10.12 |
| | D | D | 2.46 | 7.33 | 4.90 | 9.67±4.54 | 8.71±4.33 | 9.19±4.43 | 9.50 | 4.35 | 6.93 |
| | I | I | 26.02 | 16.33 | 21.18 | 6.12±6.95 | 5.13±2.91 | 5.65±3.25 | 4.75 | 6.96 | 5.85 |
| ViT [19] | R | R | 16.64 | 17.17 | 16.90 | 34.74±5.44 | 13.67±3.25 | 24.20±2.34 | 4.30 | 4.35 | 4.32 |
| | D | D | 4.30 | 3.72 | 4.01 | 8.41±5.36 | 3.83±3.76 | 6.12±2.91 | 5.66 | 0.00 | 2.83 |
| | I | I | 7.15 | 9.33 | 8.44 | 7.90±6.53 | 2.50±2.65 | 5.20±3.74 | 2.94 | 1.74 | 2.34 |
| | | | | | | Flexible modal testing | | | | | |
| FM-ViT [19] | R&D&I | R | 8.77 | 16.00 | 12.38 | 36.61±9.51 | 5.50±1.32 | 21.06±4.90 | 2.26 | 3.48 | 2.87 |
| | R&D&I | D | 5.14 | 1.83 | 3.49 | 2.79±0.44 | 1.71±1.13 | 2.25±0.36 | 2.04 | 2.61 | 2.32 |
| | R&D&I | I | 1.34 | 3.83 | 2.59 | 3.43±2.73 | 2.33±1.91 | 2.88±2.23 | 3.39 | 0.87 | 2.13 |
| CMS-Enhancer | R&D&I | R | 18.09 | 2.5 | 10.3 | 21.56±2.99 | 12.53±3.51 | 14.57±1.41 | 2.71 | 1.74 | 2.23 |
| | R&D&I | D | 3.35 | 3 | 3.18 | 3.5±1.95 | 2±0.75 | 2.42±0.79 | 4.3 | 0 | 2.15 |
| | R&D&I | I | 4.13 | 0.33 | 2.23 | 2.94±3.42 | 2.25±1.09 | 2.59±2.24 | 2.04 | 1.74 | 1.89 |
| FM-CLIP | R&D&I | R | 18.59 | 1.83 | 10.21 | 14.4±3.37 | 9.3±2.98 | 11.87±3.14 | 2.49 | 0.87 | 1.68 |
| | R&D&I | D | 3.21 | 2.83 | 3.02 | 4.16±2.08 | 1.08±1.01 | 2.29±0.68 | 3.85 | 0 | 1.92 |
| | R&D&I | I | 2.01 | 2 | 2.01 | 2.15±1.37 | 2±1.39 | 2.07±1.36 | 3.39 | 0 | 1.7 |

**Table 5: In the flexible modality scenario, the effectiveness of each component of FM-CLIP is verified, and ablation experiments are performed on three data sets WMCA ('seen'), SURF, and CeFA (Prot.4).**

| CMS-Enhancer | VLA | WMCA | SURF | CeFA(Prot. 4) |
|---|---|---|---|---|
| × | × | 3.16 | 9.78 | 11.84 |
| ✓ | × | 2.09 (-1.07) | 5.24 (-4.54) | 6.52 (-5.32) |
| × | ✓ | 1.89 (-1.27) | 5.58 (-4.2) | 7.11 (-4.73) |
| ✓ | ✓ | **1.77** (-1.39) | **1.47** (-8.31) | **2.99** (-8.85) |

**Table 6: Ablation experiments to verify the effectiveness of Spt.&Freq.-Extra and CMI on WMCA ('seen') data.**

| Spt.&Freq.-Extra. | CMI | APCER | BPCER | ACER |
|---|---|---|---|---|
| ✓ | × | 4.51 | 1.67 | 3.09 |
| × | ✓ | 3.82 | 1.86 | 2.84 |
| ✓ | ✓ | **3.01** | **1.16** | **2.09** |

**Table 7: Ablation experiments to verify the effectiveness of Spatial Extra. and Frequency Extra. on WMCA ('seen') data.**

| Spatial Extra. | Frequency Extra. | APCER | BPCER | ACER |
|---|---|---|---|---|
| ✓ | × | 4.81 | 2.03 | 3.42 |
| × | ✓ | 4.68 | 1.81 | 3.25 |
| ✓ | ✓ | **4.51** | **1.67** | **3.09** |

**Table 8: Ablation experiments to verify the effectiveness of Prompt Learning and LGPA on WMCA ('seen') data.**

| Prompt Learning | LGPA | APCER | BPCER | ACER |
|---|---|---|---|---|
| ✓ | × | 2.89 | 2.33 | 2.60 |
| × | ✓ | 2.24 | 1.81 | 2.10 |
| ✓ | ✓ | **2.05** | **1.73** | **1.89** |

**Table 9: Comparison of learnable parameters and Flops between the FM-CLIP framework and other methods.**

| Methods | # Parameters | FLOPs | ACER |
|---|---|---|---|
| FM-ViT [19] | 22.77 M | 3.85 G | 2.44 |
| FM-CLIP (Ours) | 5.34 M | 26.51 G | 1.76 |

respectively. For flexible modal scenario evaluations, extensive experiments demonstrate FM-CLIP's effectiveness and superior performance over SOTA results on SURF, WMCA, and CeFA datasets.

## 4.2 Ablation Study

**Effectiveness of each component in the FM-CLIP.** To verify the effectiveness of the component of FM-CLIP on dataset WMCA ('seen'), SURF, CeFA (Prot.4), we conducted ablation experiments in flexible modality scenarios as shown in Tab. 5. Specifically, CMS-Enhancer achieved ACER of 2.09, 5.24, and 6.52 on WMCA, SURF, and CeFA data respectively, compared with the baseline ACER of 3.16, 9.78, and 11.84, which were reduced by 1.07, 4.54, and 5.32. VLA achieved 1.89, 5.58, and 7.11 respectively, compared with the baseline were reduced by 1.27, 4.2, and 4.73. Finally, the combination

2.15, and 1.89 respectively. Furthermore, FM-CLIP achieves 1.68, 1.92, and 1.7 and reduces 1.19, 0.4, and 0.43 compared to FM-ViT

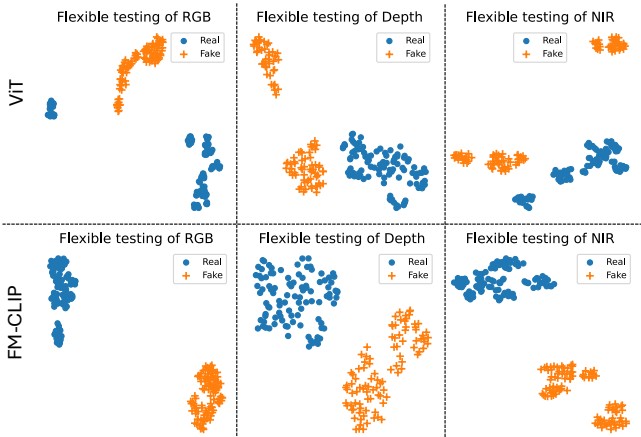

**Figure 3: Comparing the linear divisibility of visual features.**

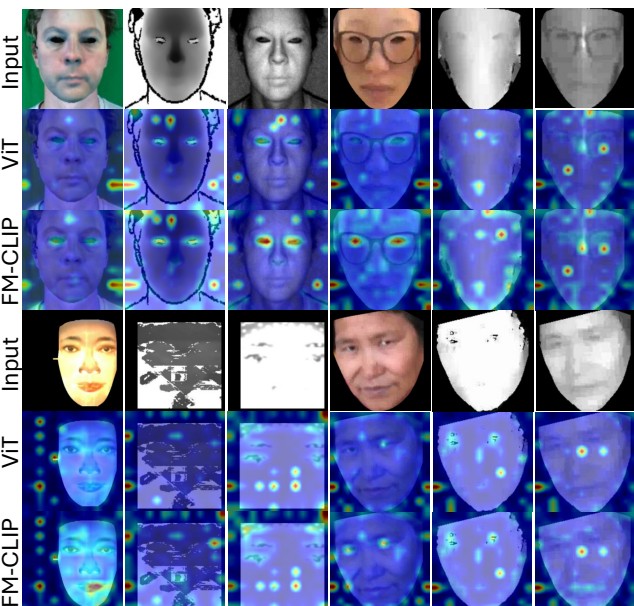

**Figure 4: Compare the distribution of feature activations.**

of CMS-Enhancer and VLA achieved 1.77, 1.47, and 2.99 for CM-CLIP, compared with the baseline were reduced by 1.39, 8.31, and 8.85 respectively.

**Effectiveness of each component in the CMS-Enhancer.** To verify the effectiveness of the component of CMS-Enhancer on dataset WMCA ('seen'), we conducted ablation experiments in flexible modality scenarios as shown in Tab. 6. Specifically, Spt.&Freq.-Extra achieved APCER, BPCER, and ACER of 4.51, 1.67, and 3.09 respectively. CMI achieved APCER, BPCER, and ACER of 3.82, 1.86 and 2.84 respectively. Finally, the combination of Spt.&Freq.-Extra and CMI achieved APCER, BPCER, and ACER of 3.01, 1.16, and 2.09 respectively. In addition to this, we also verify the performance of the spatial extractor and the frequency extractor as shown in Tab. 7. Specifically, only the spatial extractor achieved APCER, BPCER, and ACER of 4.81, 2.03, and 3.42 respectively; only the frequency extractor achieved APCER, BPCER, and ACER of 4.68, 1.81, and 3.25 respectively. Finally, the combination of the spatial extractor and frequency extractor achieved 4.51, 1.67, and 3.09 respectively.

**Effectiveness of each component in the VLA.** To verify the effectiveness of the component of VLA on dataset WMCA ('seen'), we conducted ablation experiments in flexible modality scenarios as shown in Tab. 8. Specifically, Prompt Learning achieved APCER, BPCER, and ACER of 2.89, 2.33, and 2.60 respectively; LGPA achieved APCER, BPCER, and ACER of 2.24, 1.81, and 2.10 respectively; Finally, the combination of Prompt Learning and LGPA achieved 2.05, 1.73, and 1.89 respectively.

**Model parameter analysis.** We conducted a complexity analysis on the FM-CLIP framework as shown in Tab. 9. Specifically, the FLOPs of FM-ViT [19] is 3.85, and FM-CLIP is modeled based on CLIP so FM-CLIP is 26.61. Compared with the amount of learnable parameters of FM-ViT which is 22.77M, FM-CLIP is only 5.34M mainly because we freeze the parameter updates of the visual and text backbone networks. Although the number of learnable parameters is much smaller than that of FM-ViT, the ACER of FM-CLIP on WMCA's Seen is 1.76, which is 0.68 lower than FM-ViT's 2.44, which fully demonstrates FM-CLIP's ability to handle flexible modal tasks.

**Visualization Analysis.** As shown in Fig 3, we use the UMAP tool [32] to reduce the dimensionality of the output of the features by the visual encoder to observe the linear distinguishability of categories. Compared with ViT, FM-CLIP intuitively shows that features of different categories are more compact in space and linearly separable. As shown in Fig 4, we visualized the feature activation maps of FM-CLIP on three datasets with attention-model explainability tool [2]. we verify the superiority of the proposed FM-CLIP framework from visual attention maps. Intuitively, the FM-CLIP framework not only focuses on the facial area, but also pays more attention to local facial details such as eyes, nose, and mouth. In addition, thanks to cross-modal feature enhancement, depth modality samples in faceless areas can still capture spoofing clues.

## 5 CONCLUSION

In this paper, we borrowed large-scale visual language models (VLM) to propose a novel Flexible Modal CLIP (FM-CLIP) for FAS, which can utilize text features to dynamically adjust visual features to make them independent of modality. In the vision branch, we propose a cross-modal spoofing enhancer (CMS-Enhancer), including a frequency extractor (FE) and a cross-modal interactor (CMI), aiming to map different modal attacks into a shared frequency space to perform cross-modal learning and suppress large differences between different modalities, allowing the model to effectively focus on subtle spoofing cues. In the text branch, we introduce language-guided patch alignment (LGPA) based on prompt learning, which further guides the image encoder to focus on patch-level spoofing representation through dynamic weighting of text features. Extensive experiments show that FM-CLIP is effective and outperforms SOTA methods on multiple multi-modal datasets.

## ACKNOWLEDGMENTS

This work was supported by the National Key Research and Development Plan under Grant 2021YFF0602103, the China Postdoctoral Science Foundation 2023M743756, Beijing Natural Science Foundation JQ23016, the Science and Technology Development Fund of Macau Project 0123/2022/A3, 0070/2020/AMJ, 0096/2023/RIA2, and CCF-Zhipu AI Large Model OF 202219, the Chinese National Natural Science Foundation Projects U23B2054, 62276254, and InnoHK program.

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
