# OpenReview forum: "FM-CLIP: Flexible Modal CLIP for Face Anti-Spoofing"
_acmmm.org/ACMMM/2024/Conference — MM2024 Poster_

### Official Review · Reviewer_ikEv · 2024-05-03

**Rating:** 2
**Confidence:** 4

**Summary:**

This paper presents language-guided multi-modal face anti-spoofing, which consists of two main strategies: Cross-Modal Spoofing Enhancer (CMS-Enhancer) and Language-Guided Patch Alignment (LGPA). CMS-Enhancer aims to capture spoofing clues by leveraging cross-modal learning from different modalities. LGPA aims adopt prompt learning to focus on patch-level spoofing representations. The authors claim that the proposed method is able to tackle the issue of missing modal scenario.

**Strengths:**

The authors conduct comprehensive experiments to evaluate their proposed method under fixed modal and missing-modal scenarios for multimodal face anti-spoofing.

**Limitations:**

Paper Weaknesses
1. The novelty of this paper is limited. The proposed method is composed of content from several previous methods, but the author did not cite or discuss these methods.
In Section 3.2, the concept of exploring spoof cues between RGB and frequency domains is not new and has already been proposed in a previous FAS method [A] (Please refer to Figure 2 of [A]). Next, the design Cross-Modal Interactor is highly similar to Cross-Modality Feature Fusion proposed in [B] (Please refer to Figure 2 of [B]). In Section 3.3, the proposed Language-Guided Patch Alignment is also not new. The similar idea about patch alignment learning already proposed in [C] (Please refer to Section 5 Patch Aligned Contrastive Learning of [C]). Therefore, the novelty of the proposed method appears to be incremental.
2. In Lines 389-450, the authors claim that the useful information can be selected by the uninformative gate map from other modalities. However, it is unclear why the selected information from other modalities would be useful to the current modality.
3. In Section 3.3 Vision-Language Alignment, it is unclear why the authors proposed to incorporate the text feature into the image features, as indicated by Equation (15).
4. Unclear description in experiment.
In Table 4, the authors demonstrate that the proposed method is capable of addressing the single modal scenario (e.g., only RGB or Depth). Since the proposed Cross-Modal Spoofing Enhancer extracts features from two different modalities, it remains unclear how the proposed method would operate when the input consists of a single modality.
5. The proposed method adopts a language model to guide the FAS model. The authors should discuss and compare it with recent language-guided methods, such as [D], [47], and [16].

[A] Learnable Multi-level Frequency Decomposition and Hierarchical Attention Mechanism for Generalized Face Presentation Attack Detection, WACV, 2022.

[B] Learning Polysemantic Spoof Trace: A Multi-Modal Disentanglement Network for Face Anti-spoofing, AAAI, 2023.
[C] Open Vocabulary Semantic Segmentation with Patch Aligned Contrastive Learning, CVPR, 2022.

[D] VL-FAS: Domain Generalization via Vision-Language Model For Face Anti-Spoofing, ICASSP, 2024.

[16] CFPL-FAS: Class Free Prompt Learning for Generalizable Face Anti-spoofing, CVPR, 2024.

[47] Visual Prompt Flexible-Modal Face Anti-Spoofing, TMM, 2023.

**Suitability:**

2

---

### Official Review · Reviewer_PUjC · 2024-05-07

**Rating:** 6
**Confidence:** 4

**Summary:**

This paper proposes a novel Flexible Modal CLIP (FM-CLIP), which borrows solutions from large-scale visual-language models (VLMs) to dynamically adjust visual features using text features to make them modality independent. In the vision branch, a CMS-Enhancer is proposed, including a frequency extractor (FE) and a cross-modal interactor (CMI), aiming to map attacks of different modalities into a shared frequency space to reduce interference from modality-specific signals and enhance deception cues. In the text branch, prompt-learning based LGPA is introduced to further guide the image encoder to focus on spoofing representation through dynamic weighting of text features. Through extensive experiments demonstrate the effectiveness of FM-CLIP on multiple multi-modal datasets.

**Strengths:**

1. The motivation and method design of this article are novel, and it contributes a unique solution to the flexible modal FAS task, providing reference value for the community's subsequent research.
2. This article applies the multi-modal model to the flexible-modal FAS task for the first time. This article guides the model to learn generalization features by drawing on the characteristics of CLIP multi-modal large model text features that can adaptively weight visual features.
3. The writing of the article is clear and easy to understand, and the views expressed by the author can be understood very intuitively. The article also conducted a large number of comparative experiments and ablation experiments to analyze the independent contribution of each component.
4. CMS-Enhancer is used to allow multi-modal features to interact with each other, avoiding the design of multiple backbones in traditional methods. LGPA guides visual token features through text features, and the design idea of model constraining tokens is novel.

**Limitations:**

1. When the loss is finally calculated, will all tokens in the visual features be subject to loss constraints, will it cause overfitting? Are the labels used for patch token supervision the same as the labels for CLS token features?

2. FM-CLIP may perform well on a specific data set, but its performance on unseen data sets, such as testing across scenarios, requires further verification.

3. For Frequency Extractor, DCT is used to obtain the frequency domain features of the image, and then convolution and sigmoid are used to multiply the original input. The paper mentioned the shared frequency domain space. How to understand the feature of different modes in the shared frequency domain space?

**Suitability:**

3

---

### Official Review · Reviewer_mqiG · 2024-05-23

**Rating:** 4
**Confidence:** 3

**Summary:**

The paper proposes a novel Flexible Modal CLIP (FM-CLIP) approach for face anti-spoofing (FAS). Unlike previous methods that suffer from modality-specific signal interference, FM-CLIP leverages text features to dynamically adjust visual features, making them modality-independent. The approach includes a Cross-Modal Spoofing Enhancer (CMS-Enhancer) and a Language-Guided Patch Alignment (LGPA) mechanism to enhance feature generalization and focus on spoofing cues. The method is validated through extensive experiments on multiple multi-modal datasets, demonstrating superior performance compared to state-of-the-art methods.

**Strengths:**

1. FM-CLIP introduces a novel way to handle multi-modal face anti-spoofing by leveraging text features to make visual features modality-independent. This is a significant departure from traditional methods.

2. The paper includes extensive experimental validation on multiple datasets (SURF, CeFA, WMCA), showing the method's effectiveness and superior performance over existing state-of-the-art techniques.

**Limitations:**

1. The proposed FM-CLIP method, especially with the inclusion of CMS-Enhancer and LGPA, introduces additional complexity and computational overhead. This could limit its practical deployment in real-time systems or environments with limited computational resources.

2. The paper lacks a detailed discussion on how the proposed method can be effectively integrated into real-world face recognition systems.

3. While ablation studies are conducted, more detailed analysis and comparison of the contributions of each component (e.g., CMS-Enhancer, LGPA) could provide deeper insights. Better understanding of each component's role would help in optimizing the model and potentially simplifying it without significant loss in performance.

**Suitability:**

2

---

### Official Review · Reviewer_7gre · 2024-05-24

**Rating:** 4
**Confidence:** 3

**Summary:**

The paper proposed a cross-modal feature enhancer implemented with adapter for FAS, as well as a learning approach based on text modality-invariant characteristics to reduce the learning of modality-specific features. The proposed method achieves nearly SOTA performance across various protocols.

**Strengths:**

- The introduction of text modality features for Face Anti-Spoofing is a novel and promising direction worth exploring.

- The method demonstrates excellent performance in both multi-modal and flexible modality FAS protocols.

**Limitations:**

- Flexible-modal Face Anti-Spoofing typically evaluates performance across various modality combinations, such as RGB+D and RGB+IR. However, this paper does not assess these combinations, focusing instead on the performance of single modalities (R, IR, D).
- Does the introduction of text modality information rely on external models? If so, might this lead to some unfairness when comparing with baseline methods? It is recommended to report the performance when only using the cross-modal feature enhancer.
- The article uses the frequency domain as the modality-shared feature space but lacks an explanation of the motivation behind this choice. Utilizing some visualization methods could help to demonstrate the validity of this approach.
- For the CMS module in the article, when training with three modalities together, is it necessary to perform feature enhancement pairwise?

**Suitability:**

2

---

### Meta-Review · Area_Chair_fJ2X · 2024-07-03

**Recommendation:** Accept (Poster)
**Confidence:** 4

**Metareview:**

The paper proposes a novel Flexible Modal CLIP (FM-CLIP) approach for face anti-spoofing (FAS). The introduction of text modality features for Face Anti-Spoofing is a novel and promising direction worth exploring. The method demonstrates excellent performance in both multi-modal and flexible modality FAS protocols. After one round of rebuttal, all reviewers agreed to accept this paper.